# The Two Sweet Sides of Janus Lectin Drive Crosslinking of Liposomes to Cancer Cells and Material Uptake

**DOI:** 10.3390/toxins13110792

**Published:** 2021-11-09

**Authors:** Lina Siukstaite, Francesca Rosato, Anna Mitrovic, Peter Fritz Müller, Katharina Kraus, Simona Notova, Anne Imberty, Winfried Römer

**Affiliations:** 1Faculty of Biology, University of Freiburg, 79104 Freiburg, Germany; anna.mitrovic@pluto.uni-freiburg.de (A.M.); peter.fritz.mueller@bioss.uni-freiburg.de (P.F.M.); k.kraus97@yahoo.de (K.K.); 2Signaling Research Centers BIOSS and CIBSS, University of Freiburg, 79104 Freiburg, Germany; 3CERMAV, CNRS, University of Grenoble Alpes, CNRS, CERMAV, 38000 Grenoble, France; simona.notova@cermav.cnrs.fr; 4Freiburg Institute for Advanced Studies (FRIAS), University of Freiburg, 79104 Freiburg, Germany

**Keywords:** chimeric carbohydrate, protein engineering, hypersialylation, giant unilamellar vesicles, drug delivery, cancer cell targeting, live-cell imaging

## Abstract

A chimeric, bispecific Janus lectin has recently been engineered with different, rationally oriented recognition sites. It can bind simultaneously to sialylated and fucosylated glycoconjugates. Because of its multivalent architecture, this lectin reaches nanomolar avidities for sialic acid and fucose. The lectin was designed to detect hypersialylation—a dysregulation in physiological glycosylation patterns, which promotes the tumor growth and progression of several cancer types. In this study, the characteristic properties of this bispecific Janus lectin were investigated on human cells by flow cytometry and confocal microscopy in order to understand the fundamentals of its interactions. We evaluated its potential in targeted drug delivery, precisely leading to the cellular uptake of liposomal content in human epithelial cancer cells. We successfully demonstrated that Janus lectin mediates crosslinking of glyco-decorated giant unilamellar vesicles (GUVs) and H1299 lung epithelial cells. Strikingly, the Janus lectin induced the internalization of liposomal lipids and also of complete GUVs. Our findings serve as a solid proof of concept for lectin-mediated targeted drug delivery using glyco-decorated liposomes as possible drug carriers to cells of interest. The use of Janus lectin for tumor recognition certainly broadens the possibilities for engineering diverse tailor-made lectin constructs, specifically targeting extracellular structures of high significance in pathological conditions.

## 1. Introduction

Lectins are proteins that recognize and bind carbohydrate moieties attached to proteins and lipids [1,2,3,4]. They can interact with mono- or oligosaccharides through non-covalent interactions involving hydrogen bonds, van der Waals, and hydrophobic forces. Lectins are present in all organisms: first described in plants, multiple lectins were isolated from microorganisms and animals in subsequent years [5,6]. Their interactions with carbohydrates, which can be highly specific and reversible [7,8], are crucial for numerous inter- and intracellular processes mediated by unique glycosylation patterns at the cell surface [6,9]. Thus, lectins play a significant role in biological processes such as embryonic development, cell growth and differentiation, endocytosis and molecular trafficking, as well as immunomodulation, inflammation, metastasis, and apoptosis [2,6,10,11,12]. Pathogens, in turn, use lectins to recognize and attach to carbohydrates present on host tissue, an essential process for invasion and infection [13,14,15]. 

The carbohydrate specificity of lectins is the underlying basis for their applications in biological and therapeutic research. Indeed, lectins have been investigated for a variety of new medical approaches, including cancer diagnosis, imaging, drug delivery, and treatment [16,17]. In the past few decades, it has been observed that there are precise distinctions in glycan structure and composition between physiological and pathological states. In cancer, cell surface alterations in glycan synthesis and expression have been widely described [18,19]. These aberrant glycosylation patterns are often associated with malignant transformation, development of metastasis, and more aggressive phenotypes [20]. Such carbohydrate structures are known as tumor-associated carbohydrate antigens (TACAs) and constitute promising targets for lectin-based therapy.

Interestingly, several bacterial lectins have found application in tumor detection or treatment. Shiga toxin (Stx), produced by *Shigella dysenteriae* and enterohemorrhagic *Escherichia coli* (EHEC) strains, has been explored to target human tumors in which the glycosphingolipid globotriaosylceramide (also known as Gb3) is overexpressed (Appendix A). The selective recognition of Gb3 by the B-subunit of the Shiga toxin (StxB) has raised interest in using this molecule as a tool for tumor imaging or as a carrier for cytotoxic compounds to cancer cells [21,22]. Similarly, the cholera toxin (Ctx) from *Vibrio cholerae* is an effective protein in tumor targeting. The specific affinity for the ganglioside GM1, which is highly expressed in the blood–brain barrier, resulted in the investigation of its B-subunit (CtxB) for new anti-glioma chemotherapy strategies [23] or as a sensitive retrograde neuronal tracer [24,25,26,27]. Furthermore, a novel nanoparticle formulation exploited wheat germ agglutinin (WGA) binding to its glycan receptor for local delivery of paclitaxel to the lung [28]. The receptor-mediated cellular uptake of WGA-functionalized PEG nanoparticles has been proven to be more efficient and led to a superior in vitro anticancer activity compared to the clinical paclitaxel formulation [29]. However, lectins may induce some degrees of toxicity in host cells, either indirectly by being associated with a cell-damaging, catalytic domain (e.g., Ctx, Stx, MytiLec), or directly through their ability to agglutinate red blood cells, or by affecting membrane structure (e.g., MOA) [30,31]. The toxicity is mainly related to the multivalency of lectins. This potential toxicity, which can be an asset when targeting (cancer) cells, has to be taken into account when designing lectin-based vectorization strategies.

Among the frequently occurring dysregulations in glycosylation patterns, sialylation represents an established hallmark of several cancer types, including lung, breast, ovarian, pancreatic, and prostate cancer [32]. Hypersialylation, which occurs either via (i) the upregulation of sialyltransferase enzymes; (ii) the downregulation of neuraminidase enzymes; or (iii) both [33,34], results in an accumulation of sialic acid exposed on cell surfaces. Hypersialylation promotes continuous tumor growth metastasis through several mechanisms, including enhancing immune evasion and stimulating tumor invasion and migration [32]. In clinical practice, sialylation-based cancer biomarkers such as pancreatic cancer marker CA19-9 (Sialyl-Lewis a: sLe^a^) and the sialylated mucin biomarkers CA125 (MUC16) and CA15-3 (MUC1 analog) are used to detect ovarian and breast cancers [35,36]. Different strategies have been adopted in the past few years to target these alterations in sialic acid expression, such as using chemical inhibitors of sialyltransferases and specific silencing of the gene expression of sialyltransferases [37]. Nevertheless, new treatments targeting different interplays of sialylation are continuously explored. 

A tandem construction of two carbohydrate-binding modules (CBM) from the NanI sialidase of *Clostridium perfringens* (di-CBM40) was previously described to have strong avidity for sialylated surfaces, with a Kd of 1.3 µM for 3′-siallylactose chips compared to 14.4 µM for monomeric CBM40 [38]. The high avidity for sialylated surfaces displayed by engineered multivalent CBM40 is of high interest in designing drug delivery strategies toward sialylated epitopes on cell surfaces. Recently, a chimeric bispecific Janus lectin was engineered with two rationally oriented and distinct carbohydrate recognition surfaces [39]. It was designed by assembling the sequences of the sialic acid-binding CBM40 and the fucose-binding lectin of *Ralstonia solanacearum* (RSL) (Figure 1a) and referred to as FS-Janus lectin. As RSL naturally assembles as a trimeric β-propeller fold [40], FS-Janus lectin accordingly adopts a trimeric structure comprising one RSL trimer (six fucose binding sites) linked to three CBM40 monomers (Figure 1b) [38,40]. The multivalent presentation of binding sites on each face of the FS-Janus lectin allows for simultaneous recognition and high avidity binding toward fucosylated and sialylated surfaces, with a Kd in the range of 50–100 nM [39]. The dual sugar specificity of FS-Janus lectin enabled crosslinking of heterogeneous GUV populations functionalized either with the ganglioside GM3 (exposing a terminal sialic acid) or the function-spacer-lipid construct comprised of the blood group A trisaccharide (FSL-A, with terminal fucose) linked to the phospholipid DOPE, resulting in the organization of proto-cells into modular structures resembling proto-tissues. 

In this study, we investigated the ability of FS-Janus lectin to crosslink glyco-decorated GUVs to human hypersialylated epithelial cancer cells and to induce liposomal uptake and transfer of liposomal content. A hypothetical model of FS-Janus lectin-mediated interactions is depicted in Figure 1c. Remarkably, we demonstrated that the FS-Janus lectin is a suitable tool for targeting hypersialylation and delivery of glyco-decorated liposomes to cancer cells, which opens up new possibilities in drug delivery with high specificity. 

## 2. Results

### 2.1. FS-Janus Lectin Induces Lipid Exchange between GM3- and Blood Group A Trisaccharide-Decorated GUVs

In addition to recent studies that have shown crosslinking of glyco-decorated vesicles (so-called proto-cells) and the organization of proto-cells into proto-tissues (i.e., the assembly of proto-cells into tissue-like structures) mediated by FS-Janus lectin [39], we aimed to further characterize the potential of the FS-Janus lectin. We used giant unilamellar vesicles (GUVs) (i.e., lipid bilayer spheres with variable sizes up to the dimensions of human cells that can be functionalized with diverse glycans) [41]. We incubated FS-Janus lectin (200 nM) with two populations of GUVs, functionalized with either GM3 (monosialodihexosylganglioside, containing sialic acid) or FSL-A (function-spacer-lipid comprised of the blood group A trisaccharide containing fucose) and monitored the evolving interactions over time. Vesicles were labeled by fluorescent lipids with an Atto 647N fluorophore, linked to the phospholipid DOPE (red color) and an Atto 488 fluorophore (green color) for GM3- and FSL-A-containing GUV populations, respectively. We first observed crosslinking and proto-tissue formation of two distinct populations of sialylated and fucosylated liposomes accompanied by clearly visible shape changes of liposomes, particularly at the interfaces of liposomes (Figure 2, Appendix A). Having a closer look at the zoomed areas of Figure 2, it became apparent that after 55 min in zoom 1 and after 107 min in zoom 2, the lipids of the two liposome populations mixed over time. Membrane areas of liposomes that were not forming interfaces and that were initially stained only in red became partially green, and vice versa. Thus, a lipid exchange triggered by FS-Janus lectin can be observed as indicated by yellow arrows between two differently glyco-decorated populations of GUVs. However, neither crosslinking, shape changes, nor lipid exchange were observed when the two liposome populations were incubated without FS-Janus lectin (Figure 2, control, Appendix A).

Moreover, when RSL and di-CBM40 were incubated as separate, non-chimeric components of FS-Janus lectin, we did not observe crosslinking between fucosylated and sialylated populations of GUVs (Appendix A). The obtained results showed that FS-Janus lectin can crosslink liposomes and mediate lipid exchange between sialylated and fucosylated GUVs. These results inspired us to further test the properties of FS-Janus lectin on living human cancer cells. 

### 2.2. FS-Janus Lectin and di-CBM40 Bind and Internalize into Human Epithelial Cancer Cells

As a first step to monitor the potential of FS-Janus lectin to target cancer cells and trigger the exchange of liposomal content, we evaluated its functionality in binding to human cancer cells via flow cytometry and confocal imaging. Recent studies showed that modifications in α2,6-sialylation are associated with lung cancer progression. The invasiveness and tumorigenicity of non-small cell lung cancer (NSCLC) might be mediated by alterations in the expression of sialyltransferases, as shown in A549 and H1299 NSCLC cells in vitro [42]. Thus, the human NSCLC cell line H1299 was selected as the preferred cell line for these studies. To assess the capacity of FS-Janus lectin to target sialic acid residues specifically, we compared its binding activity to the sialic acid-binding, purified di-CBM40 protein. 

The fluorescent FS-Janus lectin was incubated with H1299 cells for binding studies (Figure 3a). Three different concentrations of FS-Janus lectin (16, 32, and 64 nM) were tested to determine the optimal one. Cells were incubated with fluorescent FS-Janus lectin for 30 min on ice. Then, the unbound lectin was washed away to decrease unspecific signals. The flow cytometry analysis revealed a strong binding of the protein, probably to both fucosylated and sialylated membrane structures in a dose-dependent manner (Figure 3a). Stimulated cells were positive to all tested concentrations. The 32 nM concentration was chosen as the preferred working concentration for further binding studies by flow cytometry, as the shift in fluorescence intensity appeared remarkably distinct without reaching signal saturation. 

For comparison, the binding of the sialic acid-specific di-CBM40 to the same cancer cell line was also evaluated by flow cytometry (Figure 3b). When treated with increasing concentrations of di-CBM40, H1299 cells were efficiently stained at the plasma membrane. The binding of CBM40 to cancer cells provided a strong fluorescent signal in a range of protein concentrations from 15 to 214 nM, which also appeared to be dose-dependent (Figure 3b). 

The specific inhibition of each of the binding sites by corresponding glycans was checked by pre-incubating FS-Janus lectin with different concentrations of soluble L-fucose or 3′-sialyllactose. The concentration range of L-fucose for the complete inhibition of the RSL domain was tested before with RSL alone by flow cytometry (Appendix A). 

H1299 cells were then treated with fluorescent FS-Janus lectin in the presence of 25, 50, and 100 mM of L-fucose for 30 min on ice. After washing away the unbound lectin, a strong signal was recorded by flow cytometry (Figure 3c), confirming the proper functionality and predicted affinity of the CBM40_NanI domain for sialylated receptors. 

Next, cells were imaged with confocal microscopy to investigate the cellular uptake of FS-Janus lectin. For these experiments, the concentration of FS-Janus lectin was increased to 150 nM to improve image quality. Cells were pre-incubated with fluorescent FS-Janus lectin for 30 min on ice, and then the unbound lectin was washed away. This condition corresponded to 0 min at 37 °C in the experiment, resulting in plasma membrane staining by the fluorescent lectin. Subsequent incubation at 37 °C revealed uptake and intracellular trafficking of FS-Janus lectin, leading to an accumulation of FS-Janus lectin in the perinuclear area of the cells (white arrows), clearly visible at around 60 min of internalization (Figure 4a).

Similarly, H1299 were stimulated with di-CBM40 for internalization studies. For confocal imaging, a protein concentration of 214 nM was chosen. The intracellular uptake, trafficking, and accumulation of di-CBM40 in the perinuclear area (white arrows, Figure 4b) suggest that this domain could drive the cellular uptake of FS-Janus lectin by specifically targeting hypersialylated cell surface receptors. Furthermore, we investigated the uptake and intracellular accumulation of FS-Janus lectin, which was pre-incubated with 100 mM L-fucose in H1299 cells by confocal imaging (Figure 4c). Incubation at 37 °C with FS-Janus lectin revealed intracellular trafficking and perinuclear accumulation driven by the CBM40_NanI domain of FS-Janus lectin (white arrows). The same experiment was performed with di-CBM40, and soluble 3′-sialyllactose was used to saturate its sialic acid binding sites for flow cytometry analysis (Appendix A). Binding, uptake, and internalization of RSL (Appendix A) and FS-Janus lectin in the presence of the inhibitor 3′-sialyllactose (Appendix A) were then monitored. Appendix A shows the co-localization of FS-Janus lectin and the trans-Golgi network.

The observations made by flow cytometry and fluorescence imaging confirm that the FS-Janus lectin can selectively target sialylated conjugates on the surface of human epithelial lung cancer cells, and the CBM40 domain is sufficient for intracellular uptake.

### 2.3. FS-Janus Lectin Mediates Crosslinking of Liposomes with H1299 Cells and the Cellular Uptake of Liposomal Content

To test the ability of FS-Janus lectin to bind to cellular and synthetic membrane systems simultaneously, we incubated blood group A trisaccharide-decorated GUVs and H1299 cells together with FS-Janus lectin and monitored the interactions by live-confocal microscopy. When H1299 cells and blood group A trisaccharide-decorated GUVs were incubated over two hours without FS-Janus lectin, GUVs and H1299 cells did not interact visibly (Appendix A and Appendix A). Upon treatment with 200 nM FS-Janus lectin, GUVs (red color) in proximity to H1299 cells (partially stained in blue color) started to adhere to cell surfaces. GUVs co-localized at the contact sites to cells over time, where the FS-Janus lectin (green color) was enriched, creating straight, elongated interfaces (Figure 5, white arrows, Appendix A). Therefore, we concluded that the bispecific FS-Janus lectin mediates crosslinking between liposomes and H1299 cells. The RSL domain binds to the fucosylated surface of blood group A trisaccharide-decorated GUVs and the CBM40 domains to the sialylated receptors of H1299 cells. 

In addition to FS-Janus lectin-mediated crosslinking of liposomes and cells, we observed that FS-Janus lectin was internalized into H1299 cells, accumulating mainly in the perinuclear area over time (Figure 5 and Figure 6). Strikingly, liposomal lipids could also be found in the perinuclear area (yellow arrows in both figures), colocalizing with FS-Janus lectin. At this stage, we can only speculate that liposomal content is transferred to the host cell plasma membrane, from where it may be endocytosed. This could happen along with the uptake of FS-Janus lectin shown in Figure 5. Figure 6 depicts the kinetics of uptake and the enrichment of FS-Janus lectin (red color) and fluorescent lipids (green color) in the perinuclear area over a time window of around two hours (Appendix A).

### 2.4. FS-Janus Lectin Induces the Uptake of Complete Liposomes into H1299 Cells

One of the most promising observations we made in the framework of these studies marked the complete uptake of intact FSL-A GUVs into H1299 cells, as depicted in Figure 7 and Appendix A. Images were acquired in a live-cell imaging setting and monitored by confocal microscopy at 37 °C for 120 min. For the selected GUV, the uptake seemed to be mediated by FS-Janus lectin and took place between 25 and 30 min (indicated by yellow arrows). After the first contact of the FS-Janus lectin-bound GUV with the H1299 cell, the plasma membrane engulfed the liposome, and the filopodia-like structure of the cell appeared to drag the liposome rapidly toward the cell body (it took approximately 5 min). Even after internalization, FS-Janus lectin stood bound to the GUV. The cellular uptake of the GUV can also be nicely observed by following the displacement of the CellTrace™ Violet stain (blue color) in the cytosol by the black circular area (the liposome). As observed in Figure 4a, FS-Janus lectin alone also accumulated in the perinuclear area. 

We successfully detected liposomal uptake events multiple times for several biological replicates, using two different concentrations of FS-Janus lectin (either 200 nM or 500 nM). The uptake processes involved liposomes of different sizes and took place at various time points with varying uptake speeds. No uptake events were detected without the addition of FS-Janus lectin to glyco-decorated liposomes and H1299 cells. All these observations allowed us to conclude that the uptake of blood group A-functionalized liposomes into the cytoplasm of H1299 cells was mediated by FS-Janus lectin.

### 2.5. Uptaken Liposomes Are Deformed and Burst

Strikingly, after the cellular uptake of liposomes triggered by FS-Janus lectin (here: 500 nM), we observed liposome deformations, shrinkages, and bursts in the cytosol of H1299 cells. The experiment presented in Figure 8 and Appendix A was recorded over 180 min using a live-cell confocal microscopy setting. The liposome was internalized within the first 25 min of incubation. FS-Janus lectin (red color) remained bound to the GUV (green color), even after internalization. Additionally, in this experiment, the cellular uptake of the GUV can be nicely tracked by the movement of the black circular area in the cytosol of the cell stained by the CellTrace™ Violet marker (blue color). Around 110 min, small deformations of the liposome became apparent. In the following minutes, the liposome further deformed, and several smaller vesicular structures (nicely visible in green color) formed inside the initially internalized liposome, along with a decrease in size and partial disappearance of bound FS-Janus lectin. Between 127 and 155 min, the liposome as the entity and sub-compartments ruptured and disappeared almost completely. Whether these processes were driven by the host cell, FS-Janus lectin, or the reorganization of membrane lipids remains unclear at this stage, and further research is needed.

## 3. Discussion

We successfully demonstrated that FS-Janus lectin induces the intracellular uptake of liposomes via its dual sugar specificity. These findings provide a proof of concept for lectin-mediated targeted drug delivery using liposomes as drug carriers to cells of interest. 

Initially, this study showed that FS-Janus lectin induces crosslinking and lipid exchange between fucosylated and sialylated GUVs. In addition, the CBM40 domain, which recognizes sialic acid, drives the binding and internalization of FS-Janus lectin into H1299 cells when the RSL domain, which explicitly recognizes l-fucose, is blocked. These results inspired us to investigate the properties of FS-Janus lectin further.

As described previously, the trivalent assembly of CBM40 domains is very promising as it binds specifically to sialic acid with nanomolar avidity [38,39]. Sialic acid in cancer is an attractive target for therapeutic applications as aberrant sialylation has a significant role in cancer development. Sialoglycans such as sLe^a^, sLe^x^, STn, and GM2 are present on the surface of cancer cells and contribute extensively to cancer progression by several mechanisms, including immune system evasion and driving tumor growth and metastasis formation [43,44]. Therefore, targeting of sialic acid moieties has grown to be an innovative strategy for cancer therapy as the area for personalized medicine develops rapidly [43,45].

Janus lectin represents the concept of a scaffold with unlimited combinations of lectin modules as it can be engineered to target multiple alterations within the glycome. For instance, a well-documented example of TACA is the globoside Gb3 (also known as CD77 and P*^k^* blood group antigen), overexpressed in breast, ovarian, pancreatic, and colorectal cancer as well as in Burkitt’s lymphoma [46]. The glycosphingolipid Gb3 is an attractive therapeutic target, and lectins such as the Shiga toxin B-subunit and LecA from *P. aeruginosa* can serve for lectin-mediated tumor targeting. The replacement of the CBM40 domain in the Janus architecture with StxB or LecA represents another option to test the liposomal delivery of drugs to malignant cells of interest in the future. Furthermore, the Janus lectin approach might open new possibilities in immunotherapy, where the immune system is redirected toward tumor cells. Over the past few decades, researchers have investigated the effects of stimulation of human lymphocytes with lectins. They evaluated the activation and proliferation of B and T cells in response to lectins of different origins [47]. For instance, the Phytohemagglutinin P (PHA-P), a lectin from *Phaseolus vulgaris* (red kidney bean), possesses T cell-activating properties, which have been reported by many authors [47,48]. The idea of a Janus lectin that fuses a T cell-stimulating lectin to a TACA-specific lectin would raise novel opportunities for selective T cell-mediated cytotoxicity. 

As our results demonstrate, the FS-Janus lectin can crosslink fucose-decorated vesicles with H1299 cells and becomes enriched at the interfaces. This is encouraging for targeted drug delivery as various drugs can be loaded into vesicles. Certainly, the use of liposomes in medicine over the past few decades has offered significant prospects for effective therapies in a broad range of pathological conditions [49,50]. Lipid-based drug delivery systems have seen a remarkable increase in their application with a plethora of therapeutic compounds and diagnostic agents, described as carriers for chemotherapeutic molecules, gene therapy, and bioactive agents [50]. 

In addition to crosslinking, incubation of H1299 cells together with fucosylated vesicles and FS-Janus lectin leads to an uptake of liposomal membrane lipids. The uptaken lipids appear to colocalize with FS-Janus lectin inside the cell and accumulate in the perinuclear area over time. The internalization of smaller molecules into cells can be achieved by various endocytic mechanisms such as clathrin- or caveolin-dependent or independent endocytosis. Moreover, during vesicle uptake, we detected an elongation of the cell membrane toward the vesicles (Figure 7). This could indicate the formation of filopodia, which are known to be exhibited in H1299 cells [51]. Filopodia are actin-based membrane protrusions found in various cell types that contribute to cell migration and surface adhesion as well as endocytosis [52,53]. Macrophages, for instance, use filopodia to detect pathogens or apoptotic cells within their environment. By specific phagocytic receptors displayed on the tips of the filopodia, macrophages can bind to the surface of the pathogen or cell, which subsequently initiates a process of internalization [54]. A recent study demonstrated that filopodia on HeLa cells, a human cervical cancer cell line, serves as a route of endocytic uptake of extracellular vesicles and other nanoparticles [54]. In addition, dynamic clustering and dispersion of lipid rafts at the lamellipodia of myoblasts contributes to both cell-to-cell adhesion and plasma membrane union during the cell-to-cell fusion processes [55]. We, therefore, assumed that FS-Janus lectin bound to the membrane triggered the adhesion and uptake of GUVs.

The processes mentioned above suggest that FS-Janus lectin binding can provoke actin polymerization, leading to protrusions or filopodia formation at the plasma membrane and, subsequently, a pull-in of the liposome into H1299 cells.

Interestingly, when liposomes were functionalized with WGA, a lectin, they were only bound to adenocarcinoma human alveolar basal epithelial (A549) cells, but neither the lipid material nor liposome itself was uptaken. However, when the WGA-liposomes were incubated with oral epithelial cells (healthy), the enhanced uptake of liposomes was observed [56]. Recently, it was shown that multivalent targeting of cancer (K562) cells by core-crosslinked elastin/resilin-like polypeptide micelles led to micelle to cell crosslinking [57]. Additionally, another type of vesicle, the TAT-peptide modified vesicles, was taken up by endocytosis in ovarian carcinoma cells [58].

Strikingly, during experiments conducted with FSL-A-GUVs, we observed a burst and decay of intact vesicles inside the cytosol of cells, accompanied by shape deformation and shrinking of the respective vesicle. This process deserves further investigation, whether the burst was induced due to the nature of FS-Janus lectin alone or combined with the cell cytoskeletal pressure. However, it is known that actin dynamics can also trigger deformation of the liposome membrane [59]. Another hypothesis suggests that during intracellular trafficking, vesicles drive along the microtubules. Furthermore, the cytoskeleton is actively working on organelles, creating tension, and resulting in organelle rupture. 

We speculate that the rupture of vesicles can be caused by membrane reorganization and phase separation, responsible for the burst of uptaken liposomes in our experiments. In many cases, but not as a rule, direct cytosolic delivery of drugs loaded in liposomes is favored to prevent the degradation of pharmaceuticals in lysosomes [59].

To summarize, our findings show a prospective proof of concept for lectin-mediated uptake of lipid particles from vesicles and of intact vesicles that even burst inside cancer cells. On the other hand, it must be taken into account that sialylated residues are also exposed on healthy cells. This work does not address the interaction of FS-Janus lectin with non-malignant cells, but possible off-target interactions must be investigated in the future. Furthermore, the precise mechanism of vesicle uptake driven by the FS-Janus lectin has yet to be understood and needs to be addressed in further studies. 

Taken together, we show encouraging properties of FS-Janus lectin that promote further investigations regarding future applications, especially in the field of targeted drug delivery to cancer cells.

## 4. Conclusions

Lectins are arising in many applications in medicine as potential tools for specific targeting and drug delivery. Additionally, to recognize cells of interest, lectin–carbohydrate interactions can trigger vesicular transport into or across epithelial cells. These promising features are applied to the study of the chimeric FS-Janus lectin, which led to the delivery of vesicles to human cancer cells exhibiting hypersialylation. The concept of bifunctional Janus lectin described here is highly versatile since it can be adapted to other glycan epitopes by changing the lectins composing the scaffold. This opens a wide range of applications based on the selective targeting of the glycome in many pathological conditions.

## 5. Materials and Methods

### 5.1. Production of Lectins and Labeling

The constructs of FS-Janus lectin and CBM40 were produced in *Escherichia coli* BL21 (DE3). The detailed description of cloning, production, and purification was previously described [38,39]. Briefly, the genes of FS-Janus lectin and CBM40 were subcloned into vectors pET25(+) and pET45(+), respectively. The plasmids were transformed into *Escherichia coli* BL21 (DE3) and cultured in Luria–Bertani (LB) medium supplemented with 100 μg mL^−1^ ampicillin at 37 °C with agitation (180 rpm). Upon reaching OD_600_ of 0.6–0.8, the culture was induced by IPTG and transferred at 16 °C for 20 h with agitation (180 rpm). The cells were harvested by centrifugation and disrupted by a cell disruption system. After the centrifugation of the cell lysate, the supernatant was loaded on affinity columns, i.e., mannose agarose for FS-Janus lectin or nickel sepharose affinity for CBM40. The proteins were dialyzed, and the purity was verified by SDS-PAGE gel electrophoresis.

Lectins were dissolved at 1 mg/mL in Dulbecco’s phosphate-buffered saline (PBS) and stored at 4 °C prior usages. For fluorescent labeling, NHS-ester conjugated Alexa Fluor 488, Atto488, Alexa Fluor 647, Atto 488 DOPE, or Atto 647N DOPE (Thermo Fisher Scientific Inc., Rockford, IL, USA) were used. Fluorescent dyes were dissolved at a final concentration of 1 mg/mL in water-free DMSO (Carl Roth GmbH & Co. KG, Karlsruhe, Germany), aliquoted, and stored at −20 °C before usage. For the labeling reaction, 200 µL of lectin (1 mg/mL) was supplemented with 20 µL of a 1 M NaHCO_3_ (pH 8.5) solution. For FS-Janus lectin, the molar ratio between dye and lectin was set to 7:1, and the final ratio was 1.5:1. For di-CBM40, the molar ratio between dye and lectin was set to 3:1, and the final ratio resulted in being 0.5:1. The labeling mixture was incubated at 4 °C for 90 min, and uncoupled dyes were separated using Zeba Spin desalting columns (7k MWCO, 0.5 mL, Thermo Fisher Scientific Inc., Rockford, IL, USA). Labeled lectins were stored at 4 °C, protected from light. 

### 5.2. Composition and Preparation of GUVs

GUVs were composed of 1,2-dioleoyl-sn-glycero-3- phosphocholine (DOPC), cholesterol (both Avanti Polar Lipids, Alabaster, AL, USA), Atto 647N 1,2-dioleoyl-sn-glycero-3-phosphoethanolamine (DOPE; Sigma-Aldrich Chemie GmbH, Darmstadt, Germany), and either FSL-A(tri) (function-spacer-lipid with blood group A trisaccharide; Sigma-Aldrich Chemie GmbH, Darmstadt, Germany) and Ganglioside GM3 from bovine milk (Avanti Polar Lipids, Alabaster, AL, USA) at a molar ratio of 64.7:30:0.3:5 mol%. 

GUVs were prepared by the electroformation method as earlier described [58]. Briefly, lipids dissolved in chloroform with a total concentration of 0.5 mg/mL were spread on indium tin oxide (ITO-covered) glass slides and dried in a vacuum for at least one hour or overnight. Two ITO slides were assembled to a chamber filled with sucrose solution adapted to the osmolarity of the imaging buffer of choice, either HBSS (live-cell imaging) or PBS (GUVs only imaging). Then, an alternating electrical field with a field strength of 1 V/mm was implemented for 2.5 h at room temperature. Later, we observed the GUVs in chambers manually built as described [60]. 

### 5.3. Cell Culture

The human lung epithelial cell line H1299 (American Type Culture Collection, CRL-5803) was cultured in Roswell Park Memorial Institute (RPMI) medium supplemented with 10% fetal calf serum (FCS) and 4 mM L-glutamine at 37 °C and 5% CO_2_, under sterile conditions. Cells were cultivated in standard TC-dishes 100 (Sarstedt AG & Co. KG, Numbrecht, Germany) until 90% confluency, detached with trypsin (0.05% trypsin-EDTA solution; Sigma-Aldrich Chemie GmbH, Darmstadt, Germany) re-seeded for a subculture or for experiments. For experiments, cells were stimulated with different concentrations of FS-Janus lectin and CBM40 for indicated time points. 

### 5.4. Flow Cytometry Analysis 

H1299 cells were detached with 2 mL of 1.5 mM EDTA in PBS (-/-), and 1 × 10^5^ cells were counted and transferred to a U-bottom 96 well plate (Sarstedt AG & Co. KG, Numbrecht, Germany). To quantify protein binding to cell surface receptors, cells were incubated with different concentrations of fluorescently labeled FS-Janus lectin and CBM40 solutions for 30 min at 4 °C and protected from light compared to PBS-treated cells as a negative control. For the saturation of fucose-binding sites, 32 nM FS-Janus lectin was pre-incubated with 25, 50, or 100 mM soluble l-fucose, for 30 min at RT, in the absence of light. At the end of pre-incubation, the solution was diluted 100 times and added to cells for 30 min at 4 °C, in the dark. Subsequently, cells were centrifuged at 1600× *g* for 3 min at 4 °C and washed twice with FACS buffer (PBS (-/-) supplemented with 3% FCS *v*/*v*). After the last washing step, the cells were re-suspended with FACS buffer and transferred to FACS tubes (Kisker Biotech GmbH Co. KG, Steinfurt, Germany) on ice and protected from light. The fluorescence intensity of treated cells was monitored at FACS Gallios (Beckman Coulter Inc., Brea, CA, USA) and further analyzed using FlowJo V.10.5.3.

### 5.5. Lectin Stimulation and Fluorescence Microscopy

Between 4 and 5 × 10^4^ H1299 cells were seeded on 12-mm glass coverslips in a 4-well plate and allowed to adhere. The next day, cells were stimulated with fluorescently labeled FS-Janus or CBM40 for 30 min at 4 °C, then washed once with PBS and incubated at 37 °C for the indicated time points. Subsequently, cells were fixed with 4% paraformaldehyde for 15 min at RT. The membrane was permeabilized, and cells were blocked by 0.2% Saponin in 3% BSA in PBS (*w*/*v*) for 30 min. Nuclei were counterstained with DAPI (5 × 10^−9^ g/L), and the samples were mounted on coverslips using Mowiol (containing the anti-bleaching reagent DABCO). Samples were imaged by means of a laser scanning confocal microscope system (Nikon Eclipse Ti-E inverted microscope equipped with Nikon A1R confocal laser scanning system, 60x oil immersion objective, numerical aperture (*NA*) of 1.49, with four lasers: 405 nm, 488 nm, 561 nm, and 640 nm). The images were further analyzed using NIS-Element Confocal 4.20 from Nikon and ImageJ 1.52a from Laboratory for Optical and Computational Instrumentation. A minimum of three biological replicates with ≥20 cells per condition were analyzed.

### 5.6. Live-Cell Imaging 

H1299 cells were stained with a violet CellTrace™ proliferation kit (Thermo Fisher Scientific Inc., Waltham, MA, USA) one day prior to the experiment. A volume containing a total of 2.5 × 10^5^ cells was transferred to a 15 mL tube and centrifuged (280× *g*, 1 min). After discarding the supernatant, the cell pellet was re-suspended in 1 mL PBS, and 5 µM of violet CellTrace™ proliferation marker in DMSO was added. The solution was incubated for 20 min at 37 °C and 5% CO_2_ in the absence of light. The staining solution was diluted with 5 mL of pre-warmed RPMI medium and incubated for 5 min at 37 °C and 5% CO_2_, according to the manufacturer’s instructions. The cells were pelleted by centrifugation (280× *g*, 1 min), re-suspended in 2 mL of RPMI complete medium, and 0.5 mL of cell solution was transferred to each well of a 4-well glass-bottom dish (CELLview™; Greiner Bio-One International GmbH, Frickenhausen, Germany). The cells were incubated overnight at 37 °C and 5% CO_2_ to ensure adherence. On the day of the experiment, cells were pre-washed with and kept in Hanks’ Balanced Salt Solution (HBSS, Thermo Fisher Scientific Inc., Waltham, MA, USA) supplemented with 1% (*v*/*v*) FCS, 1 mM L-glutamine, 1% (*v*/*v*) NEAA, 10 mM HEPES, 0.55% (*w*/*v*) d-Glucose, while imaging. GUVs were pre-incubated with fluorescently labeled FS-Janus lectin for 5 min and applied to the cells with a final lectin concentration of 200 or 500 nM. GUVs and H1299 cell samples were imaged at 37 °C using an incubator stage (Okolab, Pozzuoli, Italy) mounted onto a confocal laser scanning microscope (Nikon Eclipse Ti-E, A1R). Image acquisition and processing were conducted using the software NIS-Elements (version 4.5, Nikon).

### 5.7. Chemical Reagents

The following reagents were obtained from commercial sources: RPMI 1640, PBS (-/-), FCS, and L-glutamine were all purchased from Gibco (Thermo Fisher Scientific Inc., Waltham, MA, USA). The following chemicals were obtained from Roth: BSA, DABCO, DAPI, EDTA, Mowiol, NH_4_Cl, paraformaldehyde.

## Figures and Tables

**Figure 1 toxins-13-00792-f001:**
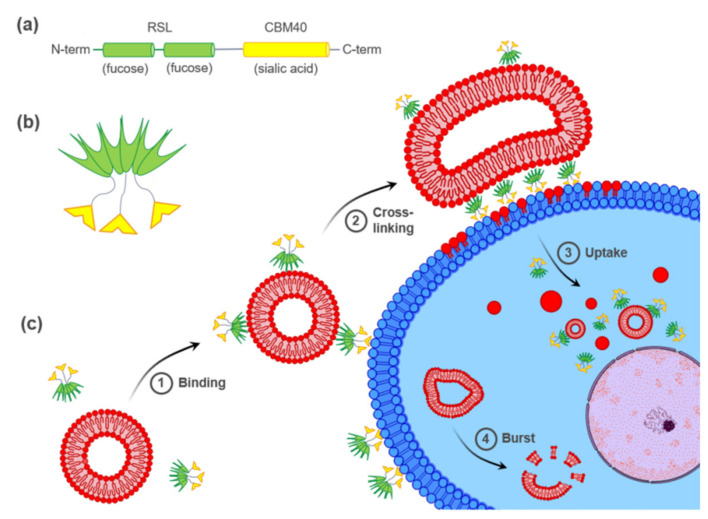
Hypothetical model of FS-Janus lectin-mediated crosslinking of glyco-decorated liposomes and human cancer cells leading to the cellular uptake of complete liposomes and liposomal content. (**a**) The peptide sequence of FS-Janus lectin with two blades of RSL (green) connected to CBM40_NanI (yellow) through a peptide linker. (**b**) Schematic representation of FS-Janus lectin presenting six fucose-binding sites at the upper face and three sialic acid-binding sites at the bottom face. (**c**) FS-Janus lectin binds simultaneously to glyco-decorated liposomes and the cell surface (1) and crosslinks them (2). These tight interactions lead to the cellular uptake of complete liposomes and liposomal content (3). Intracellular liposome transport and burst (4).

**Figure 2 toxins-13-00792-f002:**
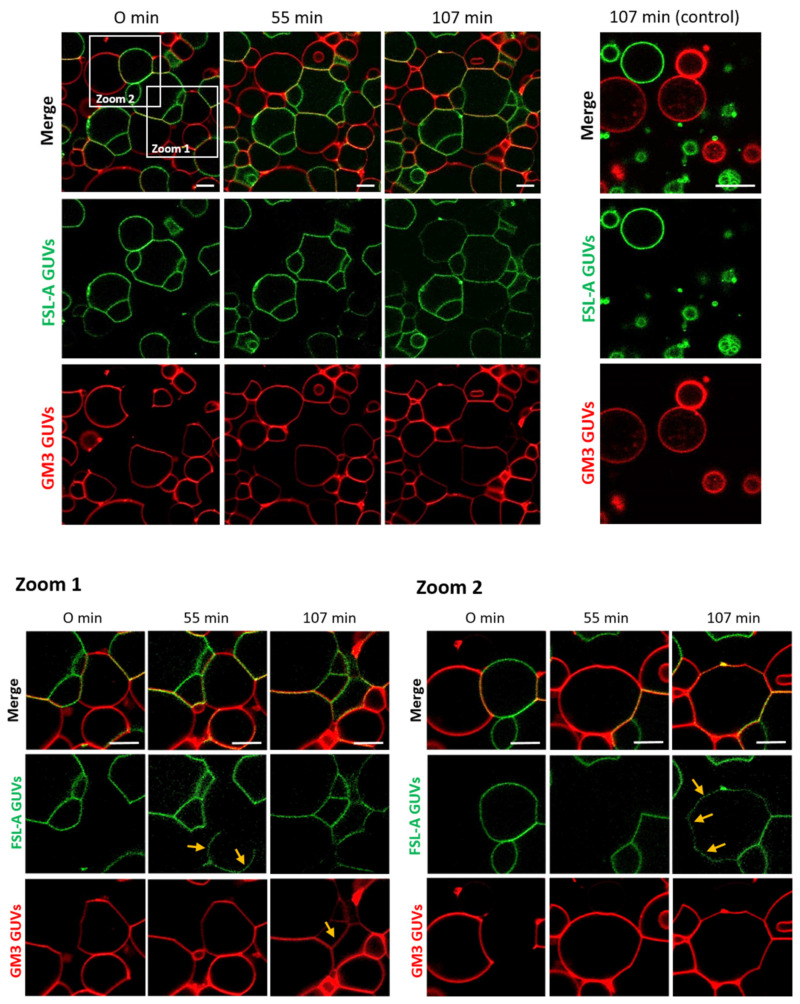
FS-Janus lectin triggers crosslinking and lipid exchange between GM3- and blood group A trisaccharide- (FSL-A) functionalized liposomes. GM3 GUVs (red color; labeled with the fluorescent lipid DOPE-Atto 647N) and FSL-A GUVs (green color; labeled with the fluorescent lipid DOPE-Atto 488) were incubated with 200 nM FS-Janus lectin (unlabeled) at room temperature and were monitored for 120 min using confocal microscopy. The crosslinking between GUVs starts immediately after the addition of FS-Janus lectin, and lipid exchange becomes visible after 55 min of incubation. Some examples are highlighted by yellow arrows. Without the addition of FS-Janus lectin, the liposomes do not crosslink and maintain their round shape over the entire incubation time. The scale bars represent 10 μm.

**Figure 3 toxins-13-00792-f003:**
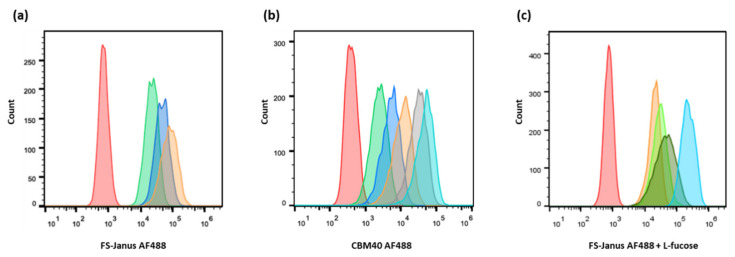
Dose-dependent binding of FS-Janus lectin and di-CBM40 to H1299 lung epithelial cells. Flow cytometry analysis of gated living H1299 cells incubated for 30 min on ice with FS-Janus lectin (**a**), di-CBM40 (**b**), and FS-Janus lectin in the presence of soluble l-fucose (**c**). (**a**) Histogram of fluorescence intensity of H1299 cells stimulated with different concentrations of FS-Janus lectin AF488 (red: negative control, green: 16 nM, blue: 32 nM, orange: 64 nM). (**b**) Histogram of fluorescence intensity of H1299 cells stimulated with different concentrations of di-CBM40 AF488 (red: negative control, green: 15 nM, blue: 21 nM, orange: 42 nM, grey: 107 nM, light blue: 214 nM). (**c**) Histogram of fluorescence intensity of H1299 cells stimulated with FS-Janus lectin AF488 (32 nM) pre-incubated with different concentrations of soluble l-fucose for 30 min at room temperature (RT) to saturate fucose-binding sites (red: negative control, light blue: FS-Janus lectin 32 nM, dark green: FS-Janus lectin 32 nM + 25 mM L-fucose, green: FS-Janus lectin 32 nM + 50 mM L-fucose, orange: FS-Janus lectin 32 nM + 100 mM L-fucose).

**Figure 4 toxins-13-00792-f004:**
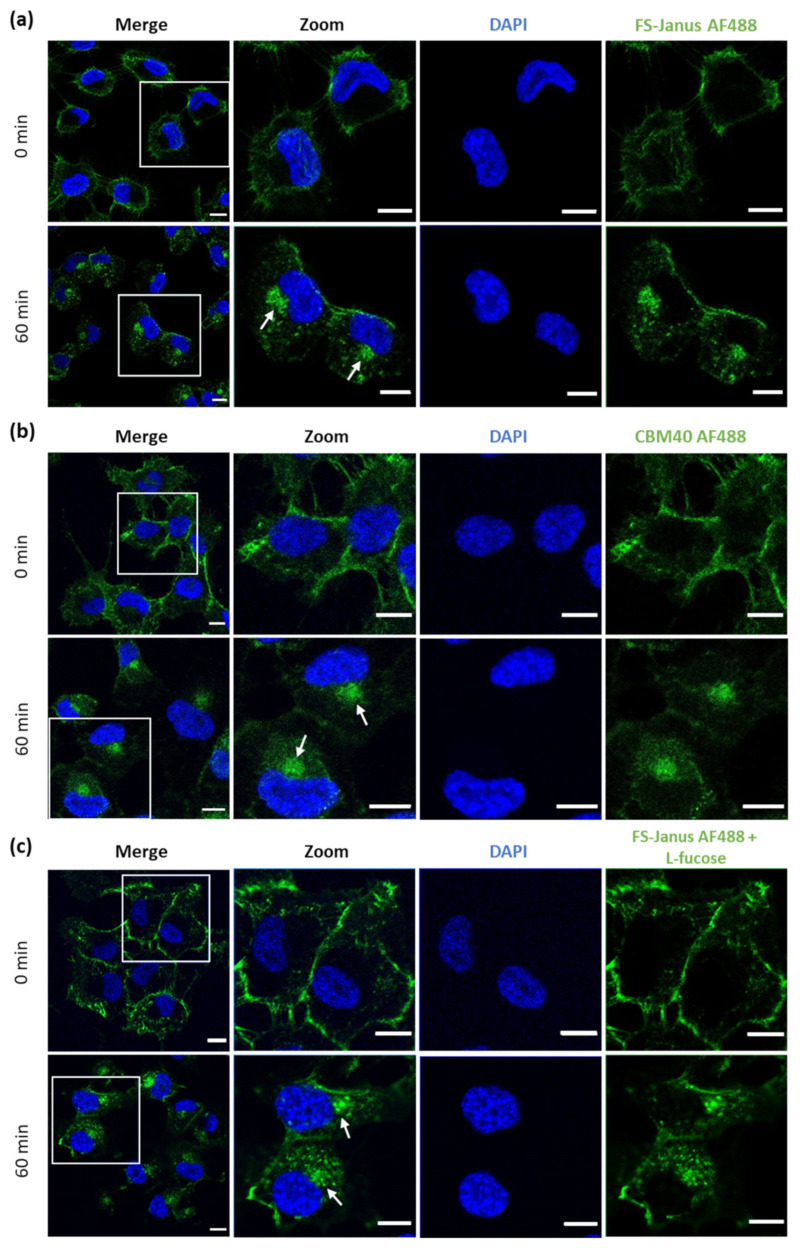
Fluorescence imaging revealed the internalization of FS-Janus lectin and di-CBM40 into H1299 cells. (**a**) Confocal imaging of fluorescently labeled FS-Janus lectin (green color) incubated with H1299 cells at different time points. FS-Janus lectin is internalized within 60 min and accumulates in the perinuclear region of H1299 cells (white arrows). Nuclei were counterstained by DAPI. (**b**) Confocal imaging of di-CBM40 (green color) binding and uptake into H1299 cells at different time points. di-CBM40 was observed intracellularly in H1299 cells after 60 min (as indicated by white arrows). (**c**) Confocal imaging of fluorescently labeled FS-Janus lectin (green color) incubated with H1299 cells at different time points. In the presence of soluble l-fucose, FS-Janus lectin was internalized within 60 min and accumulated in the perinuclear region of H1299 driven by its CBM40 domains. Nuclei were counterstained by DAPI. Scale bars represent 10 μm.

**Figure 5 toxins-13-00792-f005:**
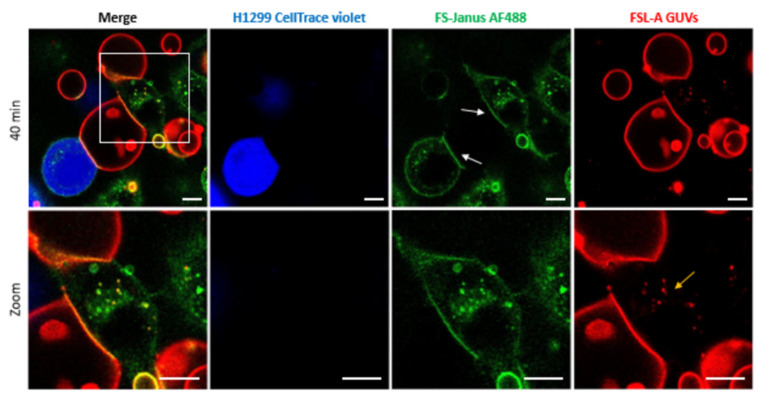
Crosslinking of blood group A trisaccharide-decorated GUVs and H1299 cells is mediated by 200 nM FS-Janus lectin. FS-Janus lectin is enriched at the interfaces (white arrows). The GUVs (red color; labeled with the fluorescent lipid DOPE-Atto 647N) were incubated with the H1299 cells (blue color; partially stained with CellTrace™ Violet) and FS-Janus lectin (green color; labeled with AF488). The yellow arrow indicates perinuclear accumulations of liposomal lipids and FS-Janus lectin. Live-cell imaging experiments were performed at 37 °C by using confocal microscopy and were recorded for 120 min. Scale bars represent 10 μm.

**Figure 6 toxins-13-00792-f006:**
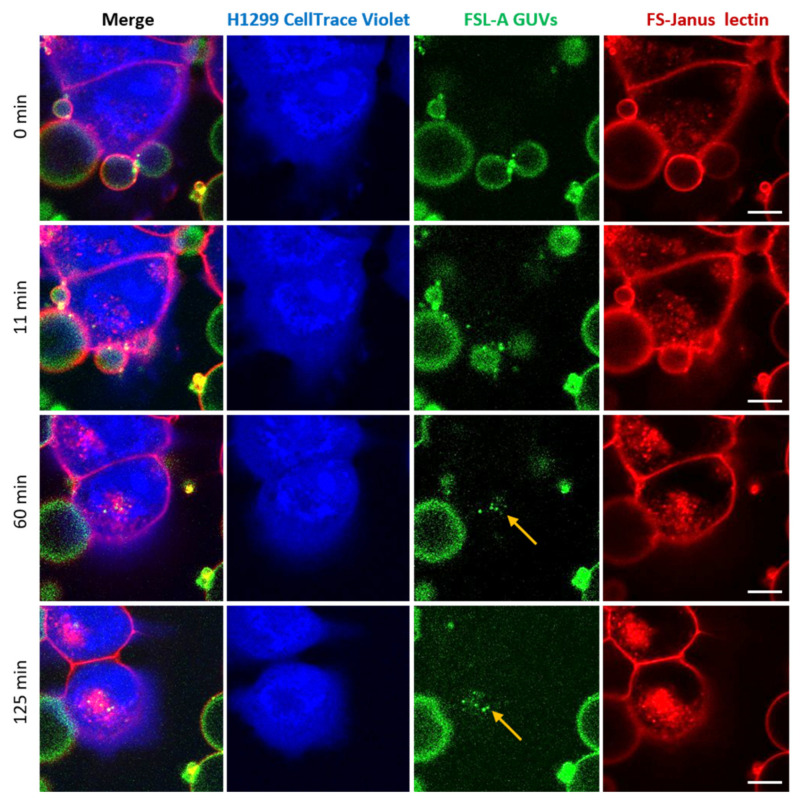
Kinetics of the cellular uptake and perinuclear accumulation of liposomal lipids and FS-Janus lectin. H1299 cells (blue color; partially stained with CellTrace™ Violet) were incubated with 200 nM FS-Janus lectin (red color, labeled with Atto 647) and blood group A trisaccharide-decorated GUVs (green color; labeled with the fluorescent lipid DOPE-Atto 488). Yellow arrows indicate perinuclear accumulation of fluorescent liposomal lipids and FS-Janus lectin. Live-cell imaging experiments were performed at 37 °C by using confocal microscopy, and the kinetics of cellular uptake were recorded for 180 min. Scale bars represent 10 μm.

**Figure 7 toxins-13-00792-f007:**
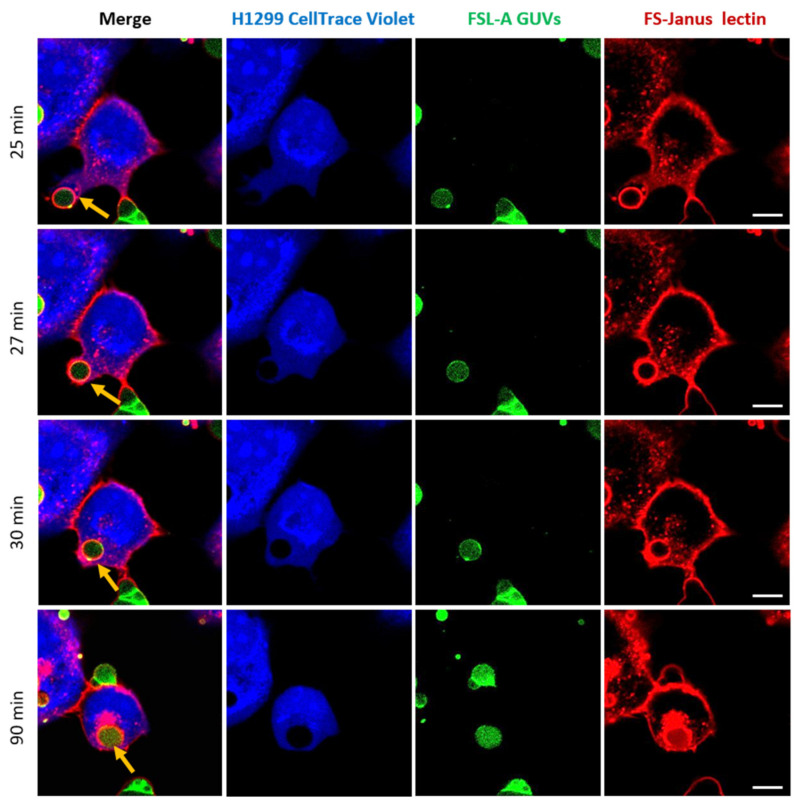
Complete uptake of blood group A-decorated GUVs into H1299 cells. FS-Janus lectin (200 nM, red color; labeled with Atto 647) triggers the internalization of complete liposomes (green color; labeled with fluorescent lipid DOPE-Atto 488) into H1299 cells (blue color; partially stained with CellTrace™ Violet). Yellow arrows point to one GUV that is taken up between 25 and 30 min. Additionally, inside the cell, the liposome remains covered by FS-Janus lectin. Live-cell imaging experiments were performed at 37 °C by using confocal microscopy, and the kinetics of uptake were recorded for 120 min. Scale bars represent 10 μm.

**Figure 8 toxins-13-00792-f008:**
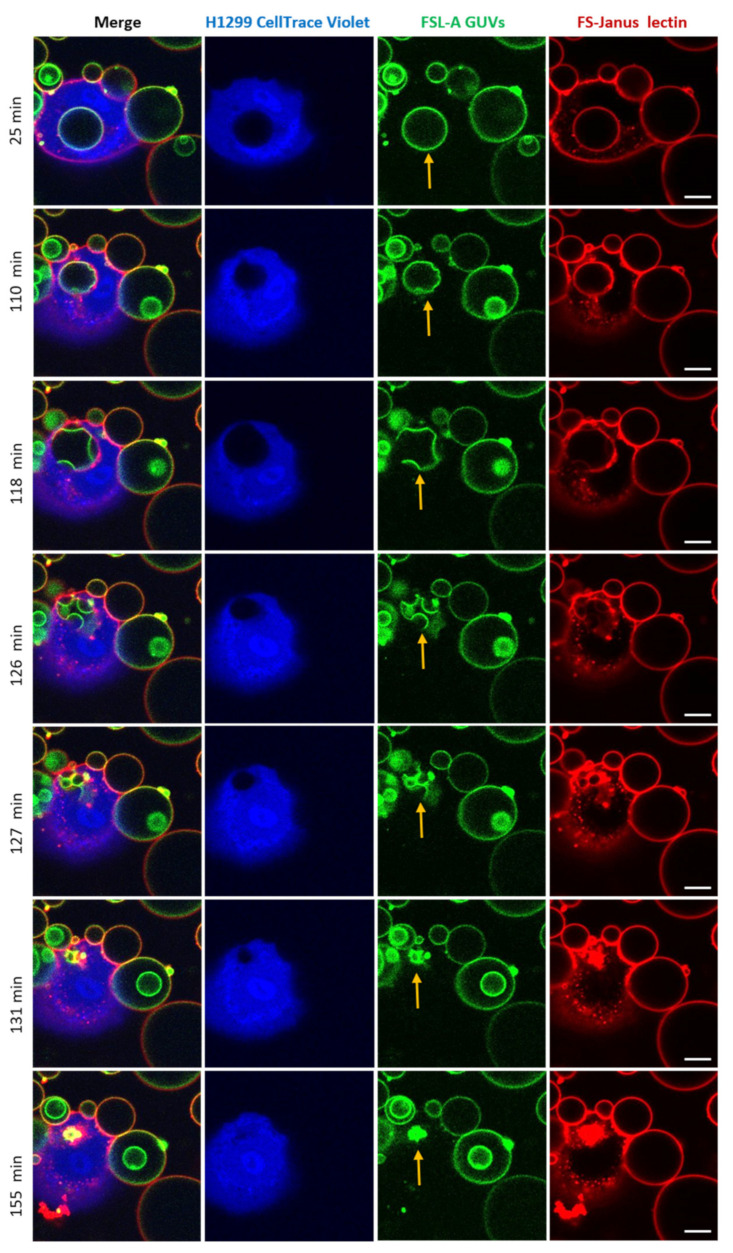
An internalized blood group A decorated GUV undergoes deformations, shrinkage, and bursts inside H1299 cells. FS-Janus lectin (500 nM, red color; labeled with Atto 647) triggered the internalization of complete liposomes (green color; labeled with fluorescent lipid DOPE-Atto 488) into H1299 cells (blue color; partially stained with CellTrace™ Violet). Once internalized, the selected GUV became deformed, reduced its size, and burst. The yellow arrows point to these events. Live-cell imaging experiments were performed at 37 °C by using confocal microscopy and were recorded for 120 min. Scale bars represent 10 μm.

## Data Availability

The datasets generated and/or analyzed during the current study are available from the corresponding authors on reasonable request.

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
