# Peer review of "The Two Sweet Sides of Janus Lectin Drive Crosslinking of Liposomes to Cancer Cells and Material Uptake"

_toxins, 2021, doi:10.3390/toxins13110792_

Round 1

Reviewer 1 Report

This is a robust study with an engineered high avidity chimeric, bispecific Janus lectin that simultaneously recognises sialylated and fucosylated glycoconjugates and readily detects hypersialylation, a dysfunction associated with tumour growth and progression of cancer. They have shown that the Janus lectin not only crosslinked fucose-labelled to hypersaliated H1299 lung epithelial cells in vitro but also facilitated the cellular uptake of the vesicles in vitro. The authors state that their findings show prospective proof of concept for lectin-mediated uptake of intact vesicles into cancer cells and that Janus-type lectins have potential uses for targeted drug delivery to cancer cells.

The principle of lectin-mediated uptake of intact vesicles is not new. Several studies have shown significant targeting and uptake advantages. However, what is unique in this study is that the constructed lectin can simultaneously recognise fucosyl residues on a unilamellar vesicle and hypersialyation residues on a cancer cell and promote cross-linking of the two. This presents an original approach, although I am unsure how it would work in vivo.

A downside of lectin-mediating targeting against cancer is that any cell, cancerous or otherwise, that expresses even low levels of the targeted glycoconjugate can be detected and potentially adversely affected in the long term. Hypersialylation is an over-expression phenomenon that occurs in many cancer cells but sialyate residues do occur on healthy cells. Does the Janus-type lectin interact with any non-cancerous cells or is its affinity for cancer cells so high that this is not a significant issue. The authors should give data on the variability of cross-linking/uptake evident with other cells or discuss this matter in the text.

Author Response

We thank the editor and all reviewers for the time they spent to improve our manuscript.

We carried out all changes as requested by reviewer 1. Please find our answers to the questions raised by reviewer 1 attached as .docx file. 

The reviewer´s comments are displayed in black, while our answers are marked in light blue.

Reviewer 2 Report

I have uploaded a pdf version with comments throughout, i refer the authors to this

The authors, have here, described the use of a chimeric lectin protein that can bind two different glycan targets while connected through a linker. While there is some amount of interest with respect to the delivery of drugs, this was actually a bit under explored in this study. It seems like a bit of a missed opportunity to not pre-load the GUVs to test the delivery for a drug, for example, to test this hypothesis. 

Generally speaking, I am uncomfortable with the use of 'crosslinking' used throughout. Crosslinking, implies that the chimera in this case covalently attaches to the targets, which may be confusing to the readers. I would suggest considering an alternative. 

  • Firstly, and my main issue with this article are some of the assertions that are, in this manuscript are currently unfounded. Figure 2, suggests that the chimera is sufficient for the binding and fusion of two different membranes when compared to liposomes only (control). This is a poorly conceived set of controls used to demonstrate such a claim. One must, do these experiment using the non-fused versions of the lectins otherwise it is not appropriate to make this claim. Furthermore, if this observation is then demonstrated to be consistent, with what is presently written, a molecular basis for this observation must be proposed, which is not found in the discussion. This, also does not relate to the later sections using cells to observe the uptake of liposomes as this is like comparing apples to oranges. The direct comparison, cannot be made. 
  • Figure 3, while I am confident that your result may be correct, it would be more convincing with a perinuclear staining performed to show actual co-localization. 

Author Response

We thank the editor and all reviewers for the time they spent to improve our manuscript.

We carried out all changes as requested by reviewer 2. Please find our answers to the questions raised by reviewer 2 attached as .docx file. 

The reviewer´s comments are displayed in black, while our answers are marked in light blue.

Reviewer 3 Report

In the article entitled “The two sweet sides of Janus lectin drive crosslinking of liposomes to cancer cells and material uptake”, the author(s) investigate the interactions between liposomes functionalized with the FS-Janus lectin and human lung epithelial cancer cell H1299.

The Introduction section presents the background of this topic, its significance, and briefly describes the basis of model of FS-Janus lectin-mediated crosslinking of functionalized liposomes to cancer cells. Within this context, I would like to recommend the authors include a few sentences on the toxic nature of some lectins. This will not only better connect the topic to this particular journal but will also prompt scientists to address the potential toxicity of this lectin in future studies.

To achieve their scientific objectives the author(s) utilized electroformation to prepare fluorescent GUVs presenting either sialic acid or fucose as binding moieties. The assays presented by the author(s) include flow cytometry and confocal fluorescence microscopy. In this respect, the experimental work brings evidence of crosslinking and lipid exchange between fucosylated and sialylated GUVs, targeting of H1299 cells, uptake of membrane lipids, an unexpected uptake of GUVs into cells, and burst/decay of intact vesicles into the cytosol. These investigations may prove very fruitful for developing advanced delivery systems that utilize liposomes as carriers of drugs, lipids, or other physiologically relevant biomolecules.

The capacity of FS-Janus lectin to target sialic acid residues is solely investigated by employing a single cell line. Nonetheless, this binding moiety is also present on normal cells, and no control experiments are included. Since the long term objective of this research relates to developing drug delivery systems, avoiding off target interactions is essential for achieving this goal. Although the proof of concept presented in this manuscript may benefit from including such control experiments, I do not consider it mandatory. However, this issue must be addressed in the Discussion section of the manuscript as necessary future work.

The Materials and Methods section includes sufficient details to enable replication by someone else skilled in the field. A minor recommendation I would like to make to the author(s) is to provide a brief description of the procedure utilized to produce the lectin. The current version only references the procedure (refs. 36 and 37); I am certain that the readers would welcome a very brief description of what the procedure entails.

The work described in this article is scientifically sound, properly referenced, and the conclusions are supported by the experimental results. In conclusion, I consider that this manuscript is of interest to readers with various backgrounds and worthy of publication in Toxins. My suggestions for improving the content should be considered as a request for a minor revision.        

Author Response

We thank the editor and all reviewers for the time they spent to improve our manuscript.

We carried out all changes as requested by reviewer 3. Please find our answers to the questions raised by reviewer 3 attached as .docx file. 

The reviewer´s comments are displayed in black, while our answers are marked in light blue.

Round 2

Reviewer 2 Report

The authors have adequately addressed all of my major concerns as detailed by their rebuttal.

I am convinced the revised manuscript is of a greater quality with the added controls and supplemental figures. I have found no other outstanding concerns.